# A Machine Learning Based Prediction Model for the Sound Absorption Coefficient of Micro-Expanded Metal Mesh (MEMM)

**Yaw-Shyan TSAY \*** and **Chiu-Yu YEH**

Department of Architecture, National Cheng Kung University, Tainan 701, Taiwan;
arch54100@apps.arch.ncku.edu.tw
**\*** Correspondence: tsayys@mail.ncku.edu.tw; Tel.: +886-6-2757575#54155

**Abstract:** Recently, micro-perforated panels (MPP) have become a popular sound absorbing material in the field of architectural acoustics. However, the cost of MPP is still high for the commercial market in Taiwan, and MPP is still not very popular compared to other sound absorbing materials and devices. The objective of this study is to develop a prediction model for MEMM via a machine learning approach. An experiment including 14 types of MEMM was first carried out in a reverberation room based on ISO 354. To predict the sound absorption coefficient of the MEMM, the capability of three conventional models and three machine learning (ML) models of the supervised learning method were studied for the development of the prediction model. The results showed that in most conventional models, the sound absorption coefficient of using an equivalent perimeter had the best agreement compared with other parameters, and the root mean square error (RMSE) between prediction models and experimental data were around 0.2~0.3. However, the RMSE of all ML models was less than 0.1, and the RMSE of the gradient boost model was 0.033 in the training sets and 0.062 in the testing sets, which showed the best agreement with the experiment data.

**Keywords:** building acoustics; sound absorption coefficient; prediction models; supervised learning method

## 1. Introduction

The micro-perforated panel (MPP) has recently become a popular sound absorber in the field of noise control and building acoustics. The attractive appearance, durability, and environmental friendliness of MPP relative to the conventional porous absorbing materials made from minerals and synthetics, which present problems of indoor air quality [1]. However, the cost of MPP is still high for the commercial market in Taiwan, so MPP is not yet popular in public buildings like public transportation stations because the costs are prohibitive to the government. In this study, we developed a micro-expanded metal mesh (MEMM) absorber with a lower cost and high sound-absorbing quality.

Many studies have proposed prediction models for perforated panels and MPPs, but most of them have assumed that the shape of the perforations are circular [1–4]. However, the MEMM expanding process makes the holes on both sides uneven (Figure 1). Therefore, before using these theoretical models to predict the absorption coefficient of MEMM, we have to make assumptions regarding transforming the geometric conditions of the perforation.

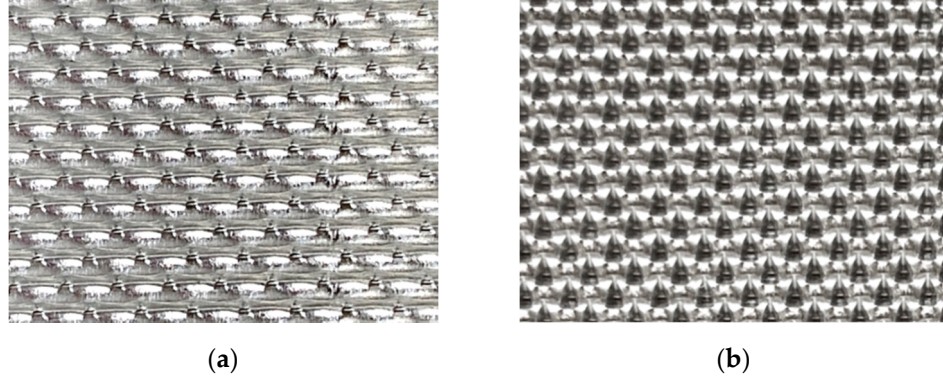

**Figure 1.** Picture of the MEMM: (**a**) Small side; (**b**) Large side.

Furthermore, machine learning (ML) can solve specific problems or perform certain tasks through relevant information and experience and has been widely used in various studies of prediction models, including image and speech recognition [5,6], market analysis [7], etc. In the ML approach, a prediction model can be trained with input data to achieve a goal without solving theoretic equations. In the field of building acoustics, ML has been adopted to predict the reverberation time (RT) of the auditorium, thus demonstrating that using neural networks has a higher correlation coefficient ($R^2$) than traditional prediction formulas, such as Sabine and Eyring [8]. Falcon Perez [9] developed the prediction model of indoor acoustic parameters in a single room; compared with the Sabine formula, the neural network had a higher mean accuracy in predicting RT. In addition, using the speech recognition method to predict RT also provided accurate prediction results [10]. Most of such ML models have achieved higher accuracy than conventional models.

The objective of this study is to develop a MEMM absorber with a lower cost that provides high sound-absorbing quality and propose a prediction model via the machine learning approach.

## 2. Materials and Methods

An experiment including 14 types of MEMM was first carried out in a reverberation room based on ISO 354. To predict the sound absorption coefficient of the MEMM, the capability of three conventional models and three machine learning models of supervised learning method were studied for the development of the prediction model.

With regard to practical use, the conventional model and the machine learning model need to input different variables (Table 1). Since we can determine the input variables of machine learning, the four variables used in this study are all simple geometric conditions that can be easily obtained. In contrast, the conventional model needs to use the microphotograph of the MEMM to convert the orifice diameter.

**Table 1.** Input variables of prediction models (O: necessary; X: unnecessary).

|  | Conventional Models | Machine Learning Models |
|---|:---:|:---:|
| **Panel thickness** | O | O |
| **Airspace depth** | O | O |
| **Perforation ratio** | O | X |
| **Orifice diameter** | O | X |
| **Coefficient of viscosity of air** | O | X |
| **Density of air** | O | X |
| **Velocity of air** | O | X |
| **Horizontal center distance of the hole** | X | O |
| **Vertical center distance of the hole** | X | O |

### 2.1. Experiment

The sound absorption performance of the MEMM was measured in the reverberation room of the architectural acoustics lab at National Cheng Kung University, Taiwan (Figures 2 and 3). The volume of the reverberation room is 171.3 m$^3$, the surface area is 184.3 m$^2$, and the floor area is 32.8 m$^2$. The laboratory uses a floating structure to reduce the outside interference of the experiment. The experiment was based on ISO 354:2003 [11], and the rating of sound absorption was based on ISO 11654 [12]. Measurements were analyzed in 1/3-octave bands with the center frequencies of 125~4000 Hz.

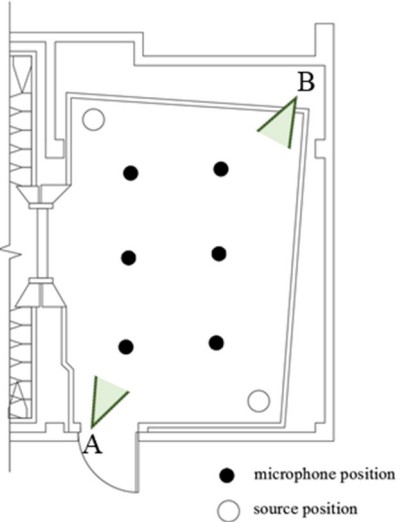

**Figure 2.** Reverberation room plan.

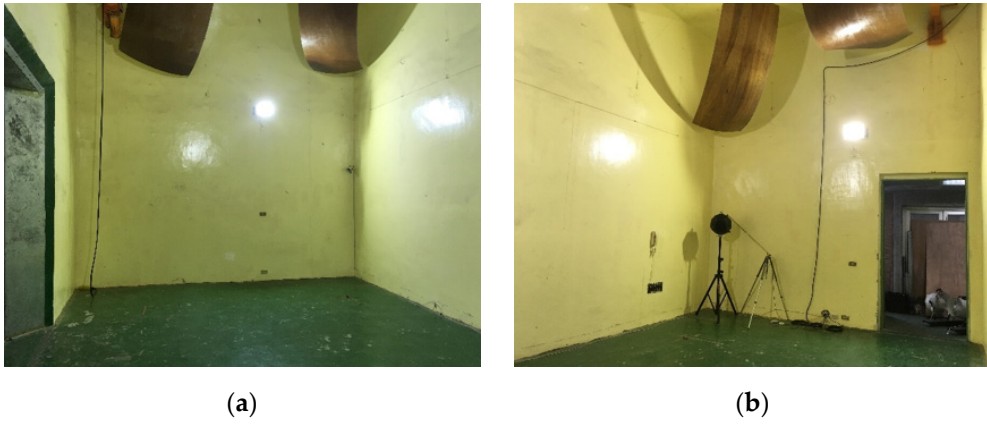

| (**a**) | (**b**) |

**Figure 3.** Picture of reverberation room: (**a**) from position A; (**b**) from position B.

The test specimen is composed of a MEMM and a closed air space layer, which together represent the structure of a common ceiling construction in Taiwan. The total area of the test specimen was 10.8 m$^2$ (3 m × 3.6 m), and each unit was 600 mm × 600 mm, and the 18 mm thick lumber core plywood is used for edge sealing (Figure 4). In the experiment, five kinds of MEMM with different hole distances and panel thicknesses were utilized with different air space depths, as shown in Table 2.

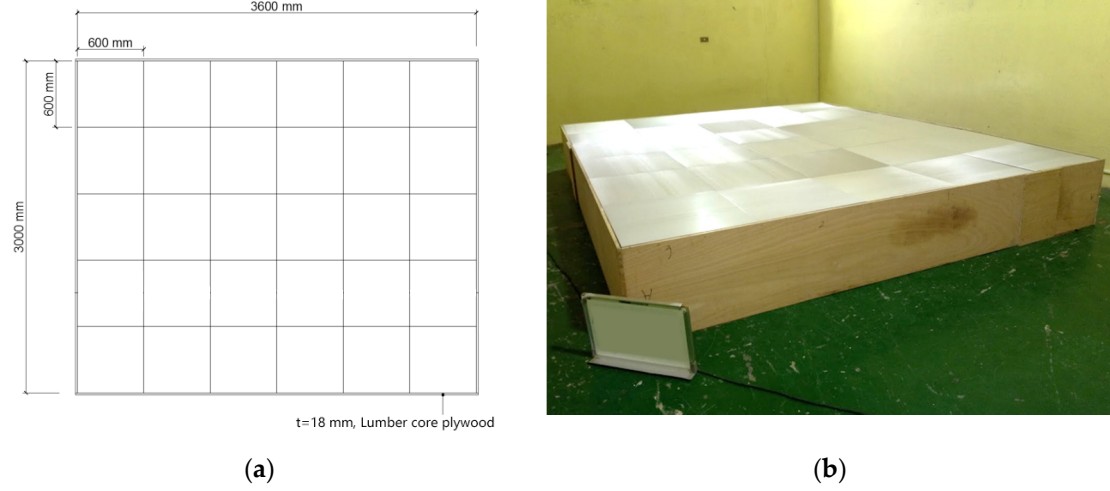

(**a**)                                                    (**b**)

**Figure 4.** Specimen of MEMM absorber: (**a**) Structure of specimen; (**b**) Picture of specimen.

**Table 2.** Experiment cases.

| Case Number | Horizontal Center Distance of the Hole (mm) | Vertical Center Distance of the Hole (mm) | Thickness of the Panel (mm) | Air Space Depth (mm) |
|---|---|---|---|---|
| A1 | | | | 210 |
| A2 | 1 | 2 | 0.5 | 260 |
| A3 | | | | 460 |
| B1 | | | | 210 |
| B2 | 2 | 4 | 0.5 | 260 |
| B3 | | | | 460 |
| C1 | | | | 210 |
| C2 | 1 | 2 | 0.6 | 260 |
| C3 | | | | 460 |
| D1 | | | | 210 |
| D2 | 2 | 4 | 0.6 | 260 |
| D3 | | | | 460 |
| E1 | | | | 200 |
| E2 | 1 | 2 | 0.8 | 450 |

## 2.2. The Conventional Models

In this study, two prediction models for the perforated panel and one for MPP were adapted for the prediction models of MEMM. Note that in these models, the assumptions of the holes are circular perforations, so the parameters of the geometric conditions of the MEMM have to be transformed to fit the model. Therefore, we adapted the circle diameters and perforation ratio obtained via equivalent area, equivalent perimeter, and the circumcircle on different sides of the panel (Figure 5, Table 3).

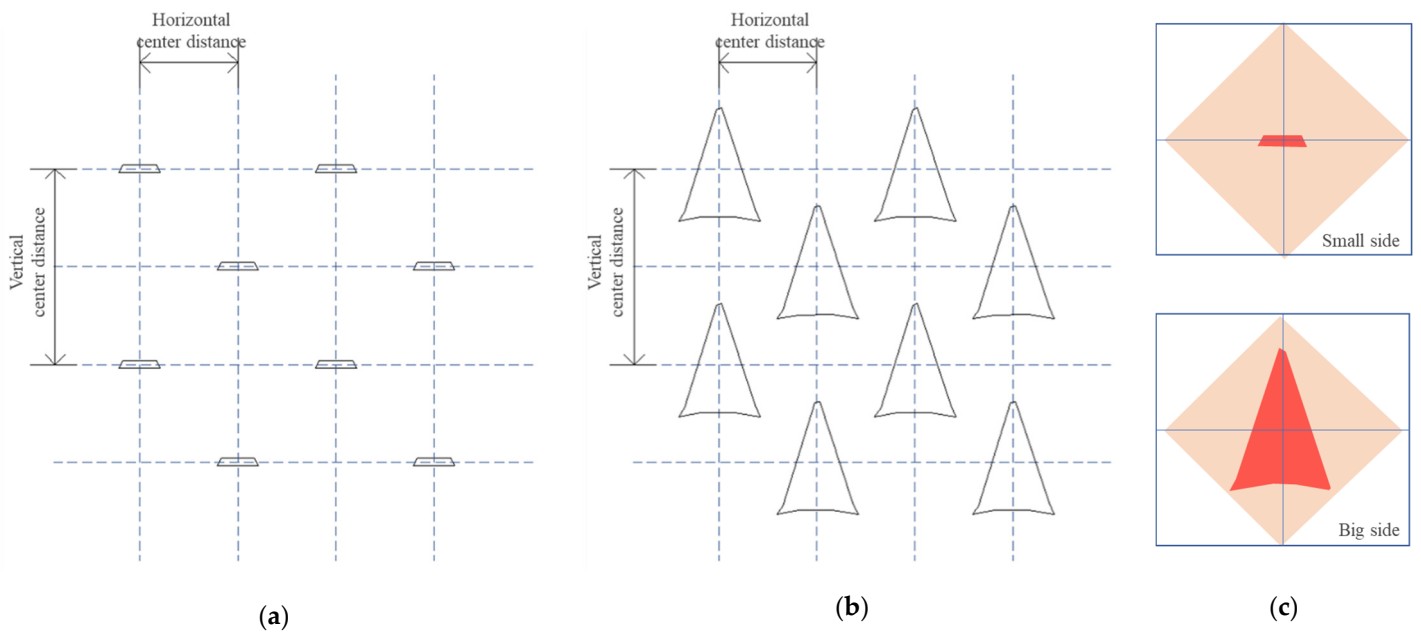

**Figure 5.** The perforation diagram of different sides: (**a**) Small side; (**b**) Large side; (**c**) Unit of each perforation.

**Table 3.** Circle orifice diameter transforming method.

| | Transforming Method | Diagram |
|---|---|---|
| Equivalent perimeter | perimeter of the hole = L $2\pi r = L$ | |
| Equivalent area | cross-sectional area of the hole = S $\pi r^2 = S$ | |
| Circumcircle | $r = R$ | |

The detailed information of perforation in the microphotograph was converted to obtain different diameters and perforation ratios. For example, when considering case E on the condition of the small side, the results are as follows: circle diameters = 0.198 mm and perforation ratio = 0.014 obtained by equivalent area; circle diameters = 0.301 mm and perforation ratio = 0.033 obtained by equivalent perimeter; circle diameters = 0.482 mm and perforation ratio = 0.084 obtained by circumcircle. Furthermore, the basic parameter settings used in these models, such as viscosity coefficient of air, density of air, and velocity of air, were set at a temperature of 25 °C.

2.2.1. The Semi-Theoretical Model of a Perforated Panel

The semi-theoretical model of a perforated panel assumes that each hole in the perforated panel and the air space layer behind it are considered Helmholtz resonances [2,13,14]. The panel thickness, orifice diameter, and distance of each hole, perforation ratio, and air space depth are represented in Equations (1)–(9).

Equations (1)–(3) show the acoustic impedance of hole $Z_h$, acoustic resistance $R$, which is based on the viscosity and heat conduction of the inner wall of the hole, and correction factor $\delta$ for the orifice diameter. Equations (4) and (5) represent the characteristic impedance $z_a$ and propagation constant $\gamma_a$ of air. Equations (6)–(8) obtain the transfer matrix of the air space layer, the transfer matrix with the perforated panel, and the acoustic impedance $Z$ of the entire sound absorber, respectively. Finally, the sound absorption coefficient $\alpha$ can be obtained from Equation (9).

$$Z_h = R + j\omega\rho_0 \frac{t + 2\delta d}{S} \text{ ,} \tag{1}$$

$$R \cong 4\frac{0.83 \times 10^{-2} \sqrt{f}}{S} \frac{t + d}{d} \text{ ,} \tag{2}$$

$$\delta = 0.4\left(1 - 1.47\sigma^{1/2} + 0.47\sigma^{3/2}\right), \tag{3}$$

$$z_a = \rho_0 c_0 \text{ ,} \tag{4}$$

$$\gamma_a = jk \text{ ,} \tag{5}$$

$$\begin{bmatrix} t_{11} & t_{12} \\ t_{21} & t_{22} \end{bmatrix} = \begin{bmatrix} \cosh\gamma D & z\sinh\gamma D \\ \frac{1}{z}\sinh\gamma D & \cosh\gamma D \end{bmatrix}, \tag{6}$$

$$\begin{bmatrix} T_{11} & T_{12} \\ T_{21} & T_{22} \end{bmatrix} = \begin{bmatrix} 1 & Z_h \\ 0 & 1 \end{bmatrix}\begin{bmatrix} t_{11} & t_{12} \\ t_{21} & t_{22} \end{bmatrix}, \tag{7}$$

$$Z = \frac{T_{11}}{T_{21}} \, , \tag{8}$$

$$\alpha = 1 - \left| \frac{Z-1}{Z+1} \right|^2 , \tag{9}$$

### 2.2.2. Lee & Kwon's Model

For the prediction model of perforated panels, a transfer matrix method is used instead of an equivalent circuit to calculate the effect of sound through the perforated panel and the air space layer [3]. The calculation considered the panel thickness, orifice diameter, perforation ratio, and air space depth.

In this model, the empirical formula of Rao and Munjal [15] was corrected via Equation (10) to calculate the normalized acoustic impedance $\xi$ of the perforated panel. Equations (11) and (12) represent the transfer matrix of the perforated panel and the air space layer, respectively. The transfer matrix of the whole sound absorber can be obtained by Equation (13). Equations (14) and (15) calculate the reflection coefficient $\gamma'$ of sound and obtain the sound absorption coefficient $\alpha$.

$$\xi = \left[ 7.337 \times 10^{-3} + j \times 1.3 \times 2.2245 \times 10^{-5} (1 + 51t)(1 + 204d) f \right] / \sigma \, , \tag{10}$$

$$\begin{bmatrix} P_{11} & P_{12} \\ P_{21} & P_{22} \end{bmatrix} = \begin{bmatrix} 1 & \rho_0 c_0 \xi \\ 0 & 1 \end{bmatrix} , \tag{11}$$

$$\begin{bmatrix} S_{11} & S_{12} \\ S_{21} & S_{22} \end{bmatrix} = \begin{bmatrix} \cos kD & (j\rho_0 c_0) \sin kD \\ (j/\rho_0 c_0) \sin kD & \cos kD \end{bmatrix} , \tag{12}$$

$$[T] = [P][S], \tag{13}$$

$$\gamma' = \frac{T_{11} - \rho_0 c_0 T_{21}}{T_{11} + \rho_0 c_0 T_{21}} \, , \tag{14}$$

$$\alpha = \frac{4 Re(1 + \gamma' / 1 - \gamma')}{[1 + Re(1 + \gamma'/1 - \gamma')]^2 + [Im(1 + \gamma'/1 - \gamma')]^2} \, , \tag{15}$$

### 2.2.3. Maa's Model for MPP

In Maa's model, the perforation of MPP was limited to less than 1 mm in diameter and 1% in perforation ratio [4,16]. The panel thickness, orifice diameter, and distance of each hole, the perforation ratio, and the air space depth are all taken into consideration in this model, as shown in Equations (16)–(20).

Equation (16) calculates the acoustic impedance $Z_{MPP}$ of the MPP; the acoustic resistance $r$ and mass reactance $\omega m$ are calculated by Equations (17) and (18), respectively. Equation (19) shows the perforation constant $k'$ for the holes of the MMP, and the sound absorption coefficient $\alpha$ can be obtained using Equation (20) through the previously calculated sound resistance and reactance.

$$Z_{MPP} = r + j\omega m \tag{16}$$

$$r = \frac{32\eta t}{\sigma \rho_0 c_0 d^2} k_r \, , \ k_r = \left[ 1 + \frac{k'^2}{32} \right]^{1/2} + \frac{\sqrt{2}}{32} k' \frac{d}{t} \, , \tag{17}$$

$$\omega m = \frac{\omega t}{\sigma c} k_m \, , \ k_m = 1 + \left[ 1 + \frac{k'^2}{2} \right]^{-1/2} + 0.85 \frac{d}{t} \, , \tag{18}$$

$$k' = d \sqrt{\omega \rho_0 / 4\eta}, \tag{19}$$

$$\alpha = \frac{4r}{(1+r)^2 + (\omega m - \cot(\omega D/c))^2} \, , \tag{20}$$

### 2.3. The Machine Learning Model

In the ML model, a supervised learning method was used to train and obtain the prediction model in this study. In the supervised learning method, the MEMM characteristics of panel thickness, hole-to-hole distance, and air space depth were defined as input objects, and the output value was the sound absorption coefficient.

The sound absorption coefficient of each frequency band was predicted via the ML process, and the results were then compared with the experiment data to verify the applicability. The root-mean-square error (RMSE) value between the prediction models and experiment data were used to evaluate applicability.

The following describes the basic process of ML. First, the data were sorted and divided into training, validation, and testing sets. In this study, the proportion was 65% training and validation set and 35% testing sets. Furthermore, the k-fold cross validation was used (with k = 5), so that the training and validation set would be shuffled.

The second step involves selecting models and adjusting the parameters of the model. In this study, we used three kinds of ensemble learning methods, including the gradient boosting (Gboost) model, the average model, and the stacking model, to obtain the prediction model. The last two models are the combination method of the Gboost model and the three linear models. The algorithms of the linear model are shown in Table 4 [17]. Note that the advantage of the ensemble learning method is that it combines multiple learners to produce more accurate results than individual learners [18].

**Table 4.** Algorithm of linear regression models.

| Model | Objective Function |
|---|---|
| Lasso | $\min\limits_{w} \frac{1}{2n_{samples}} \|Xw - y\|_2^2 + \alpha\|w\|_1$ |
| Elastic net (ENet) | $\min\limits_{w} \frac{1}{2n_{samples}} \|Xw - y\|_2^2 + \alpha\rho\|w\|_1 + \frac{\alpha(1-\rho)}{2}\|w\|_2^2$ |
| Kernel ridge (KRR) | combines Ridge regression with the kernel trick<br>Ridge: $\min\limits_{w} \|Xw - y\|_2^2 + \alpha\|w\|_2^2$ |

1. The Gboost model

The Gboost model is a combination of the gradient descending model and the boost model. A boost model can generate a strong learner from an ensemble of weak learners, each of which can barely do better than random guessing [19]. The process of the Gboost model is to build a model, then increase the weight of data that is incorrectly predicted on this model to create a second model, and repeat the same steps to obtain a better-performing model.

2. The average model

The average model constructs and trains different models, combines the models to reduce errors and overfitting, and averages the output values of each model. In this study, the average model is a combination of the above models (three linear models and Gboost), and the average value of the four models was used as the final output value.

3. The stacking model

The problem that an averaging method may encounter is that not every model is good, and a bad model will cause the averaged result to be worse. Therefore, the stacking model is an improvement of the average model by assigning weight to the contribution of each model (in the average model, each

model contributes the same amount). The specific method uses trainable combiners, which develop a learner (another meta-model) in order to first combine the models. Doing so allows researchers to determine which learners are likely to be successful in which part of the feature space and then combine them accordingly [19]. The prediction results of the first-level model are used as the input variables of the meta-model (Figure 6).

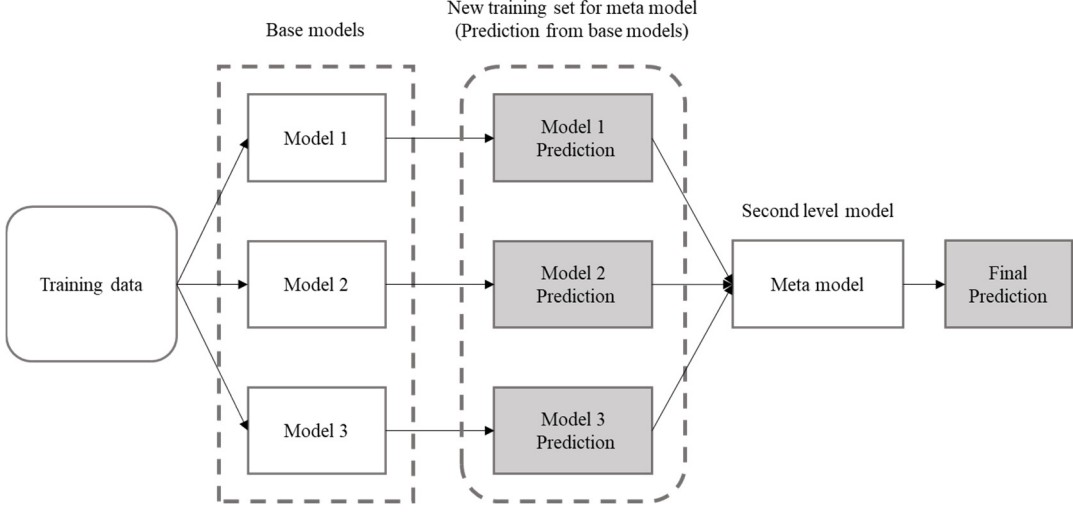

**Figure 6.** The stacking model.

## 3. Results

### 3.1. The Conventional Model

The equivalent area, equivalent perimeter, and circumcircle of the holes are used to obtain the value of the orifice diameter. Comparing the small and big sides, we found that using the geometric condition of the small side was more accurate in any transform method; therefore, the following discussion uses the results of the small side. The results of different transform methods are shown in Figure 7. Furthermore, Figure 8 shows the results of different models with the equivalent perimeter condition.

The orifice diameter converted by equivalent perimeter shows that Maa's formula had the best agreement with the experiment. However, the three models are more accurate in the low frequency band, and the error is higher in the high frequency.

Figure 9 shows that the predicted value tends to under-estimate the sound absorption coefficient of the MEMM. The root mean square error (RMSE) of the different models are listed in Table 5.

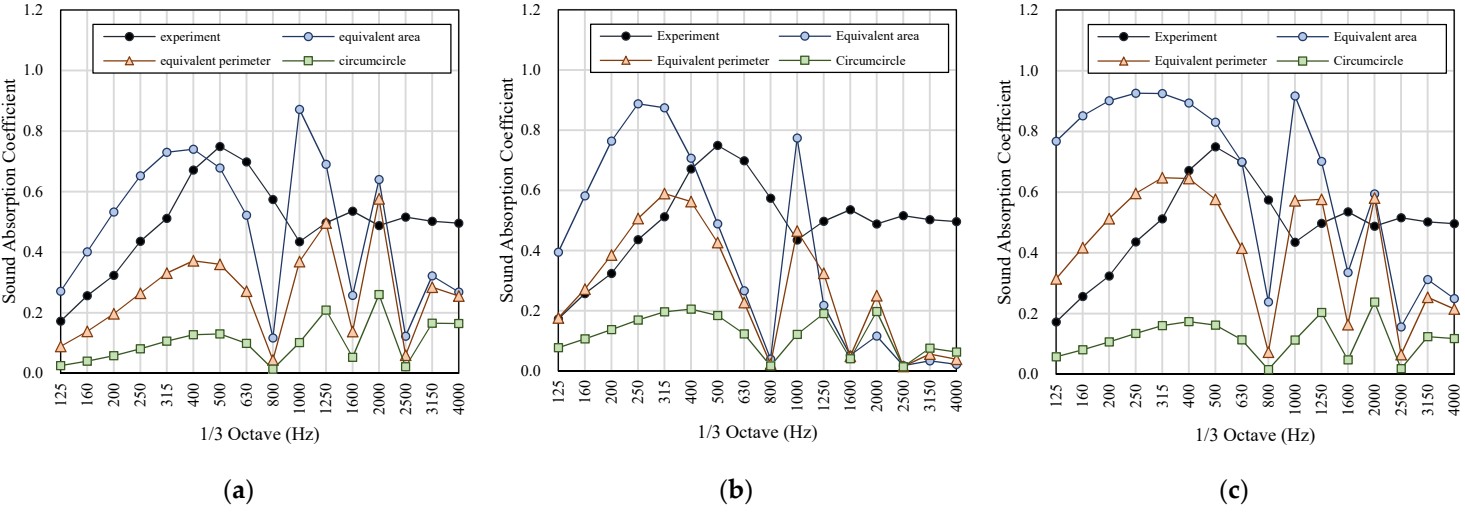

**Figure 7.** Comparison of the different methods to the converted orifice diameter (using the small side condition): case E1. (**a**) Semi-theoretical model; (**b**) Lee & Kwon's model; (**c**) Maa's model.

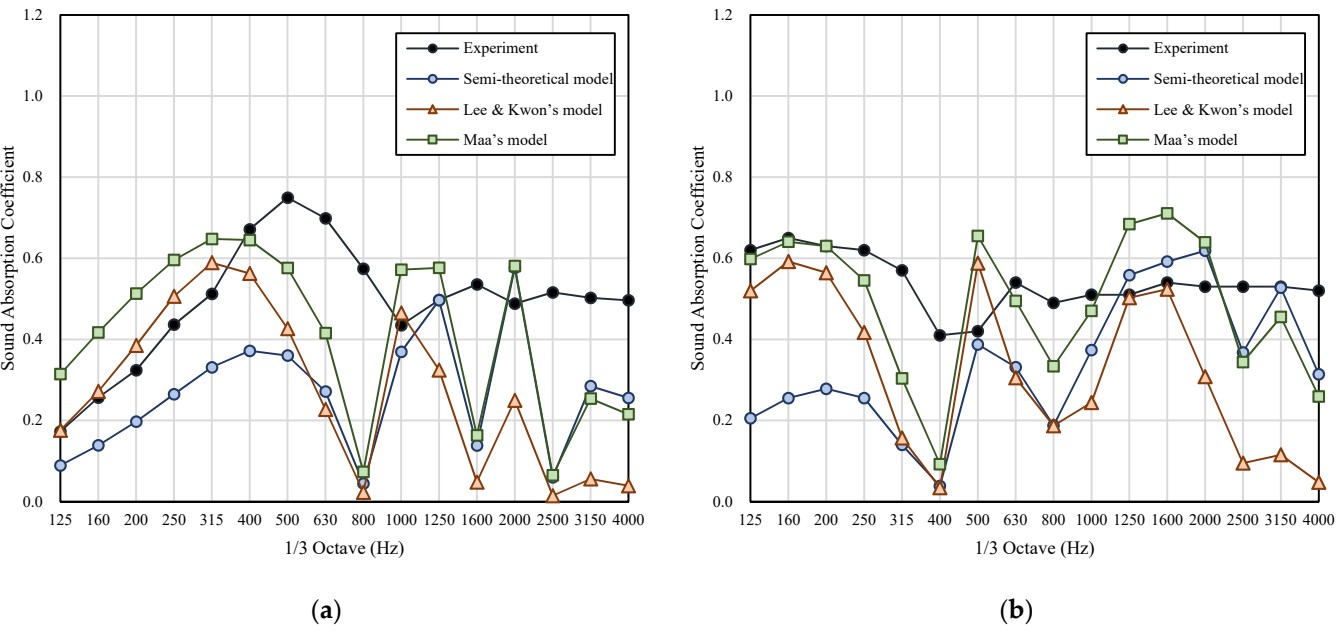

(**a**)                                                                                          (**b**)

**Figure 8.** Results of conventional models with small side equivalent perimeter condition: (**a**) case E1; (**b**) case E2.

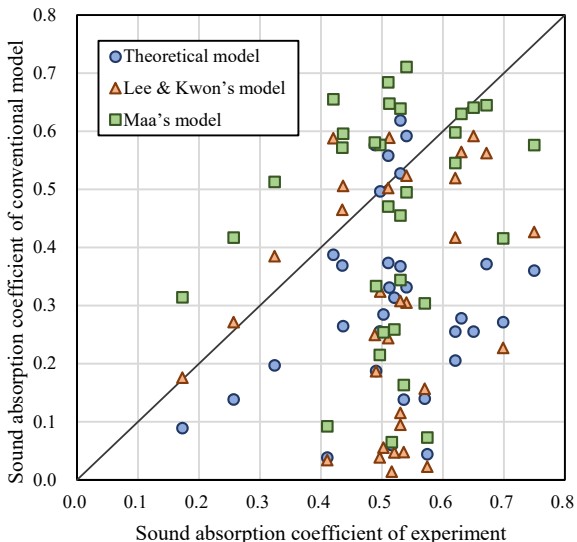

**Figure 9.** Scatter plot of conventional models.

**Table 5.** RMSE of conventional models.

| Model | Converted Method | RMSE |
|---|---|---|
| Semi-theoretical model | Equivalent area | 0.248 |
| | Equivalent perimeter | 0.276 |
| | Circumcircle | 0.413 |
| Lee & Kwon's model | Equivalent area | 0.394 |
| | Equivalent perimeter | 0.301 |
| | Circumcircle | 0.400 |
| Maa's model | Equivalent area | 0.368 |
| | Equivalent perimeter | 0.212 |
| | Circumcircle | 0.401 |

*3.2. The ML Model*

In the machine learning part (Figures 10 and 11), the prediction of the basic linear regression models was poor and failed to express the characteristics of such resonance absorption structure performing better in a specific frequency band (resonance frequency). For the ensemble model, the Gboost model and the stacking model were more accurate. The Gboost and stacking models are shown to express better sound absorption at resonance frequency. Compared to conventional models, ML models had better predictive ability for medium and high frequencies. Nevertheless, the average model could not fit the sound absorption trend. Furthermore, the prediction performance of the training set was better than that of the test set. The nomenclature and prediction results of other cases are attached in Appendix A.

Table 6 shows the RMSE of each model's prediction value and the experimented value. The RMSE of all ML models was less than 0.1. The Gradient boost model had a RMSE of 0.033 in the training set and 0.062 in the testing set, which is superior to the conventional theoretical model and shows the best agreement with experiment data.

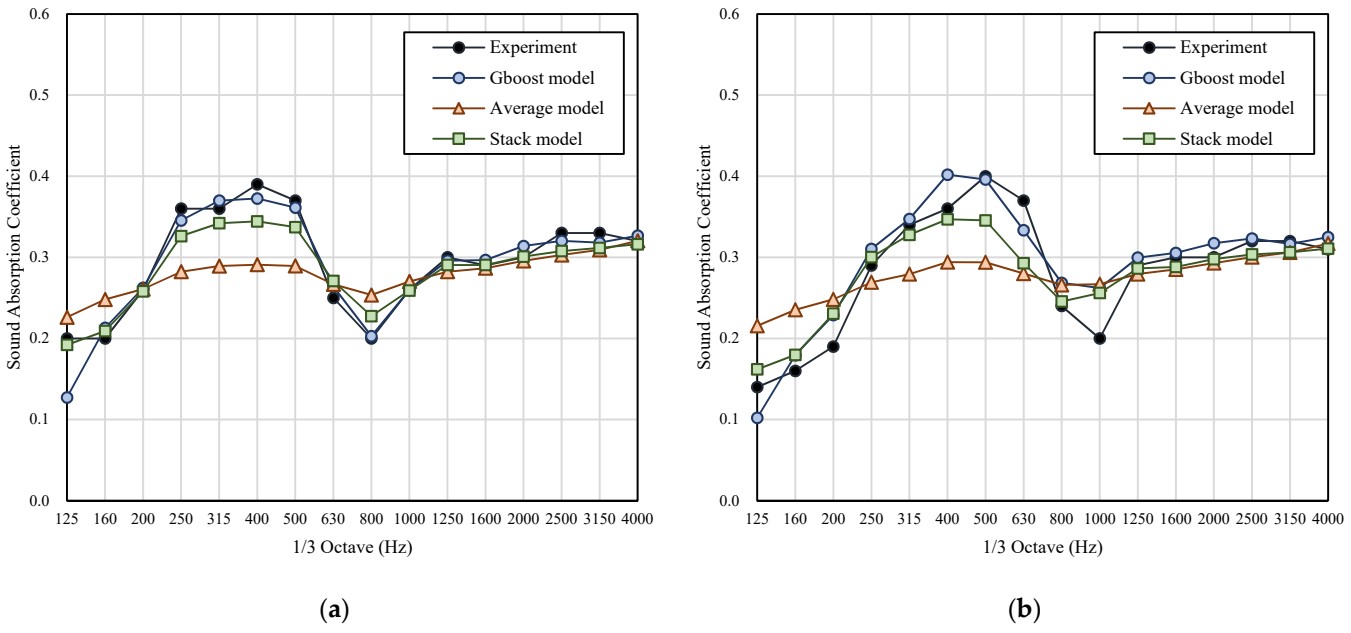

**Figure 10.** Results of machine learning models: (**a**) training set: case A2; (**b**) testing set: case A1.

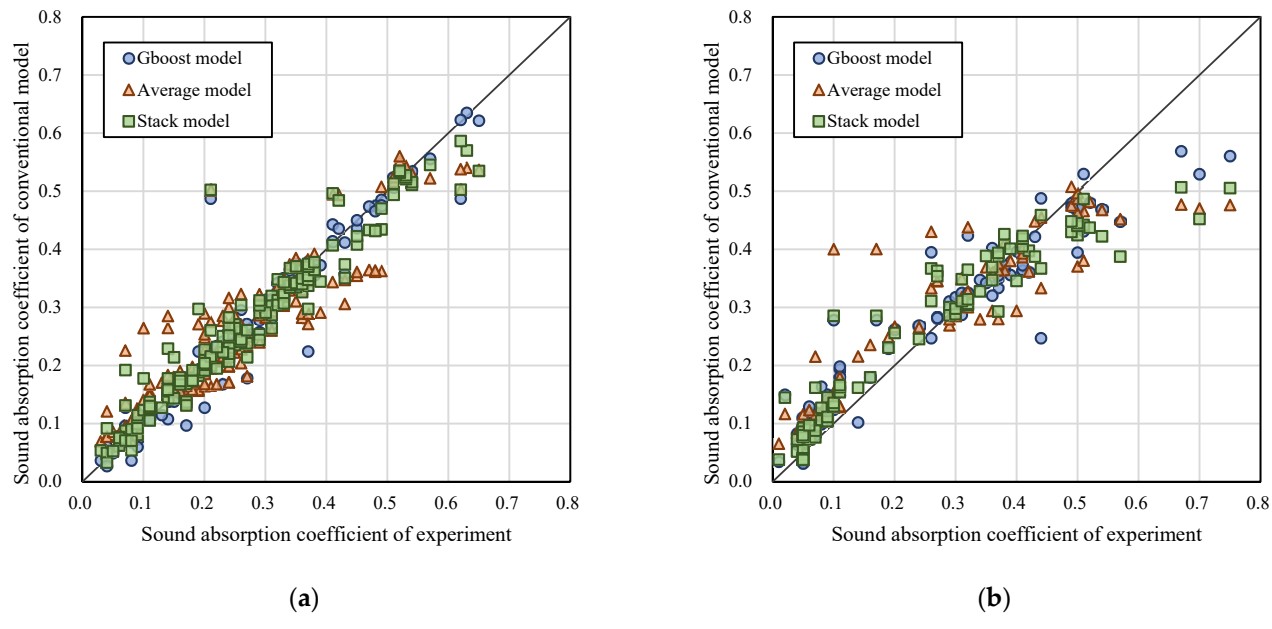

(**a**) (**b**)

**Figure 11.** Scatter plot of machine learning models: (**a**) training set; (**b**) testing set.

**Table 6.** RMSE of ML models.

|              | Model   | RMSE  |
| ------------ | ------- | ----- |
| Training set | Lasso   | 0.069 |
|              | ENet    | 0.069 |
|              | KRR     | 0.070 |
|              | Gboost  | 0.033 |
|              | Average | 0.056 |
|              | Stack   | 0.040 |
| Testing set  | Lasso   | 0.092 |
|              | ENet    | 0.092 |
|              | KRR     | 0.095 |
|              | Gboost  | 0.062 |
|              | Average | 0.081 |
|              | Stack   | 0.067 |

## 4. Discussion and Conclusions

This study attempted to use different models to predict the sound absorption coefficient of the MEMM and discuss its applicability. After comparing the conventional model and the machine learning model, the machine learning model is more accurate with regard to predictive ability. As for use, the geometric conditions of materials used in machine learning are simple to obtain, while traditional models require microphotograph and conversion methods for orifice diameter.

In the first resonance frequency band, the predictive ability of the conventional model was not poor, but regardless of the model, there was several resonance frequency bands (several resonance frequencies), which are not show in the experimental results. The assumptions of the conventional model were all speculated to be ideal physical environments. In the literature, the model was compared with the measurement of impedance tubes to demonstrate the results of several resonance frequency bands [3]. However, the results of this study, whose experiment was conducted in a reverberation room, has no such situation. The measurement method of sound absorption consists of small-size—impedance tubes [20,21] and large-size—reverberation room [11]. Further research is warranted to clarify the difference and connection between the two.

The ML model did not have this problem because the sound absorption coefficient of the training set was obtained from the reverberation room experiment. In the comparison of each ML model, the average model was usually found to underestimate the sound absorption capacity of the MEMM at the resonance frequency. It was speculated that during the average process, the result was affected by the poorly performing linear model. Furthermore, the predictive ability still differed between the training set and the testing set. If the number of the entire data set is increased, the model could be further improved to decrease the performance gap between the two.

The generalization of machine learning models primarily depends on the data set. Therefore, its predictive ability is more credible within the scope of the experiment of this study. For other types of MEMM, their applicability remains to be studied. Furthermore, if the number and diversity of experimental specimen are increased in the future, the scope of application of the prediction models can be continuously increased.

**Author Contributions:** Conceptualization, Y.-S.T.; methodology, Y.-S.T. and C.-Y.Y.; software, C.-Y.Y.; validation, Y.-S.T. and C.-Y.Y.; data curation, Y.-S.T. and C.-Y.Y.; writing—original draft preparation, C.-Y.Y.; writing—review and editing, Y.-S.T.; visualization, C.-Y.Y.; supervision, Y.-S.T. All authors have read and agreed to the published version of the manuscript.

**Funding:** This research received no external funding.

**Conflicts of Interest:** The authors declare no conflict of interest.

## Nomenclature

| | |
|---|---|
| $Z_h$ | acoustic impedance of the hole (Pa·s/m$^3$) |
| $R$ | acoustic resistance based on the viscosity and heat conduction of the inner wall of the hole (Pa·s/m$^3$) |
| $f$ | the frequency (Hz) |
| $\omega$ | the angular frequency (rad/s) |
| $\rho_0$ | the density of air (kg/m$^3$) |
| $t$ | the panel thickness (m) |
| $\delta$ | correction factor (-) |
| $d$ | the orifice diameter (m) |
| $S = \pi d^2/4$ | cross section area of the hole (m$^2$) |
| $\sigma$ | the perforation ratio (-) |
| $z_a$ | characteristic impedance of air (Pa·s/m$^3$) |
| $\gamma_a$ | propagation constant of air (rad/m) |
| $c_0$ | the velocity of air (m/s) |
| $k$ | the wave number (rad/m) |
| $D$ | cavity thickness/airspace depth (m) |
| $Z$ | acoustic impedance of the absorber (Pa·s/m$^3$) |
| $\alpha$ | the absorption coefficient (-) |
| $\xi$ | the normalized acoustic impedance of the panel (Pa·s/m$^3$) |
| $[]$ | The overall transfer matrix for perforated panel system |
| $\gamma'$ | the pressure reflection coefficient (-) |
| $Z_{MPP}$ | acoustic impedance of the MPP (Pa·s/m$^3$) |
| $r$ | relative acoustic resistance (Pa·s/m$^3$) |
| $x_m = \omega m$ | mass reactance (Pa·s/m$^3$) |
| $k'$ | the perforate constant |
| $k_r$ | the resistance coefficient |
| $k_m$ | mass reactance coefficient |
| $\eta$ | coefficient of viscosity (Pa·s) |

## Appendix A

*Training Set*

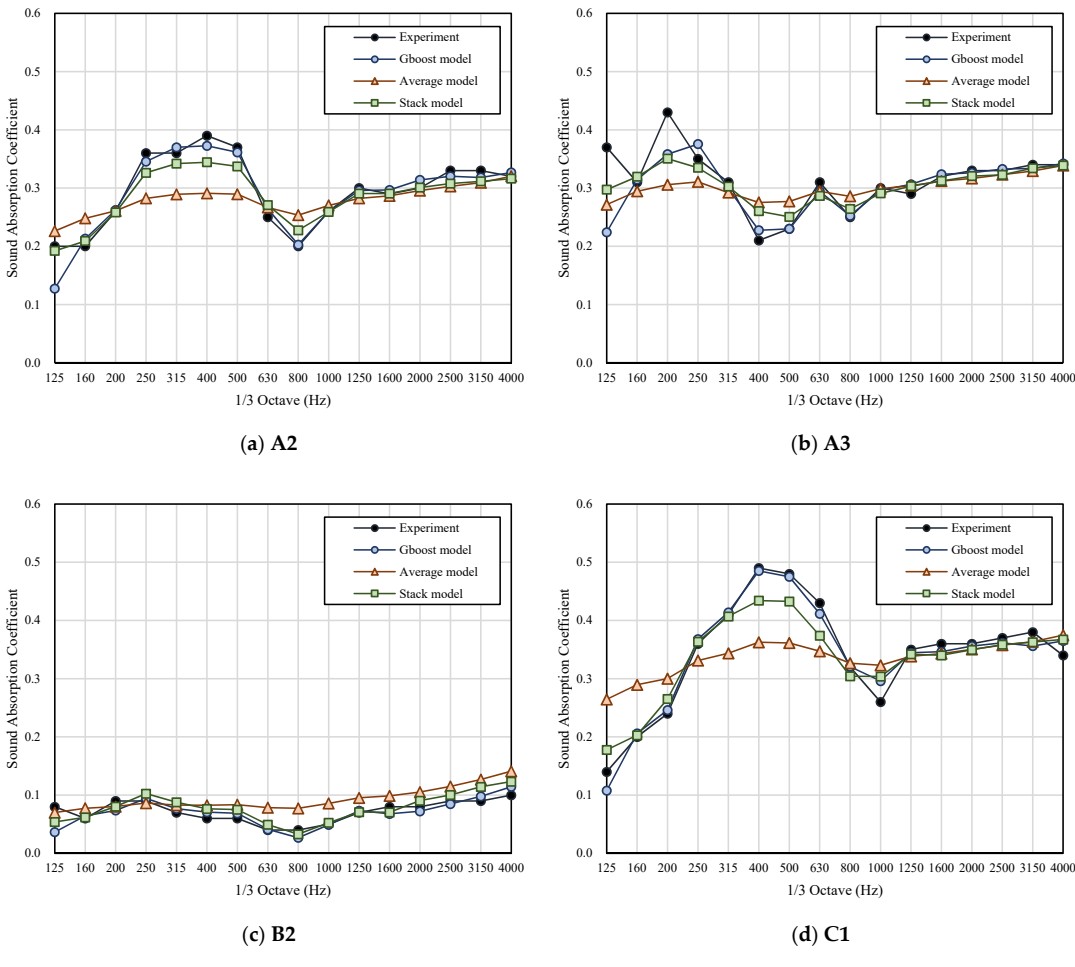

**Figure A1.** *Cont.*

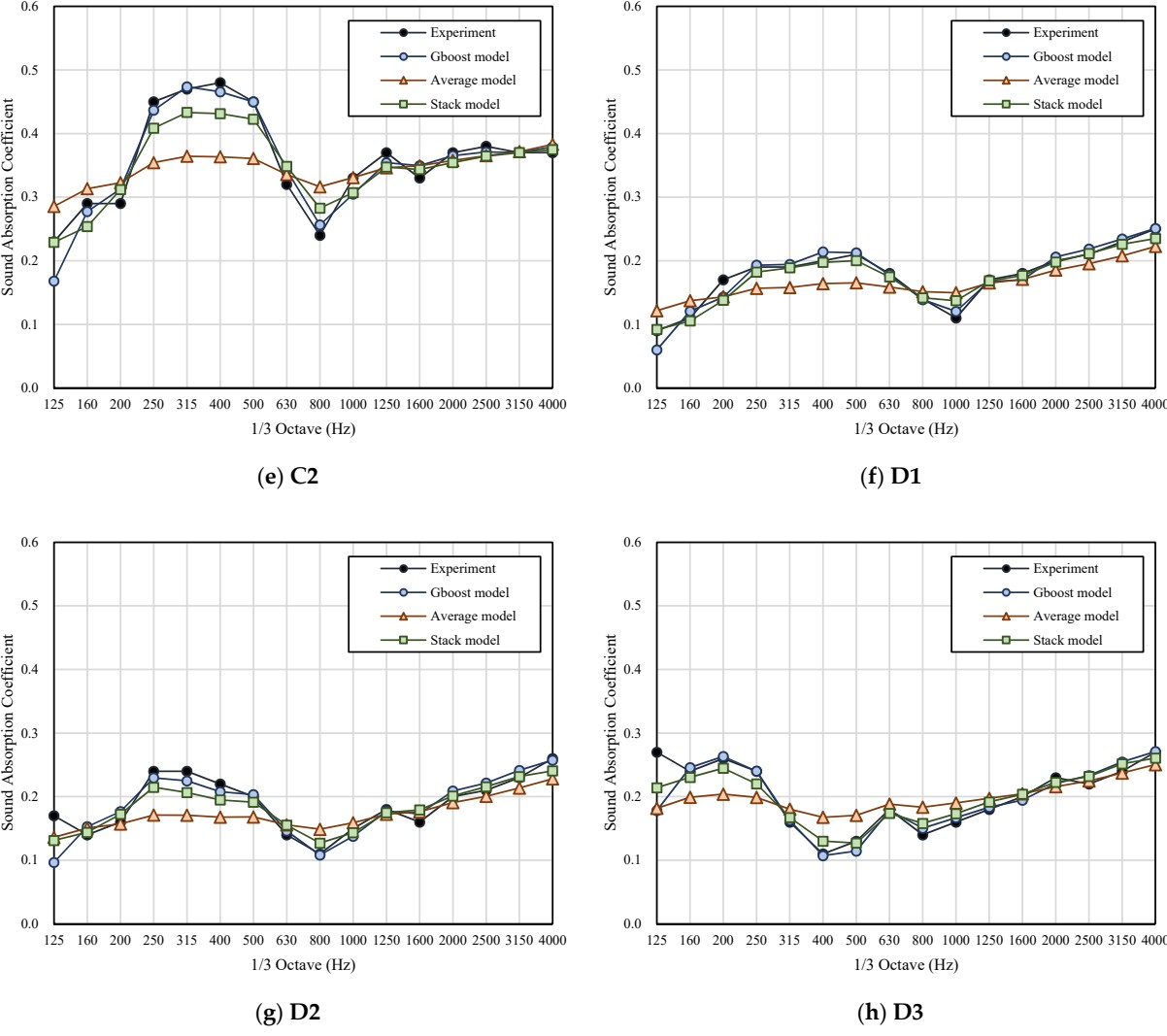

**(e) C2**

**(f) D1**

**(g) D2**

**(h) D3**

**Figure A1.** *Cont.*

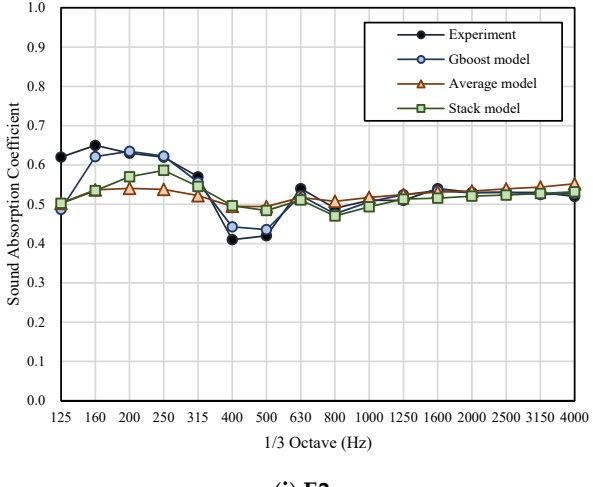

**(i) E2**

**Figure A1.** Prediction Results of the ML Model.

*Testing Set*

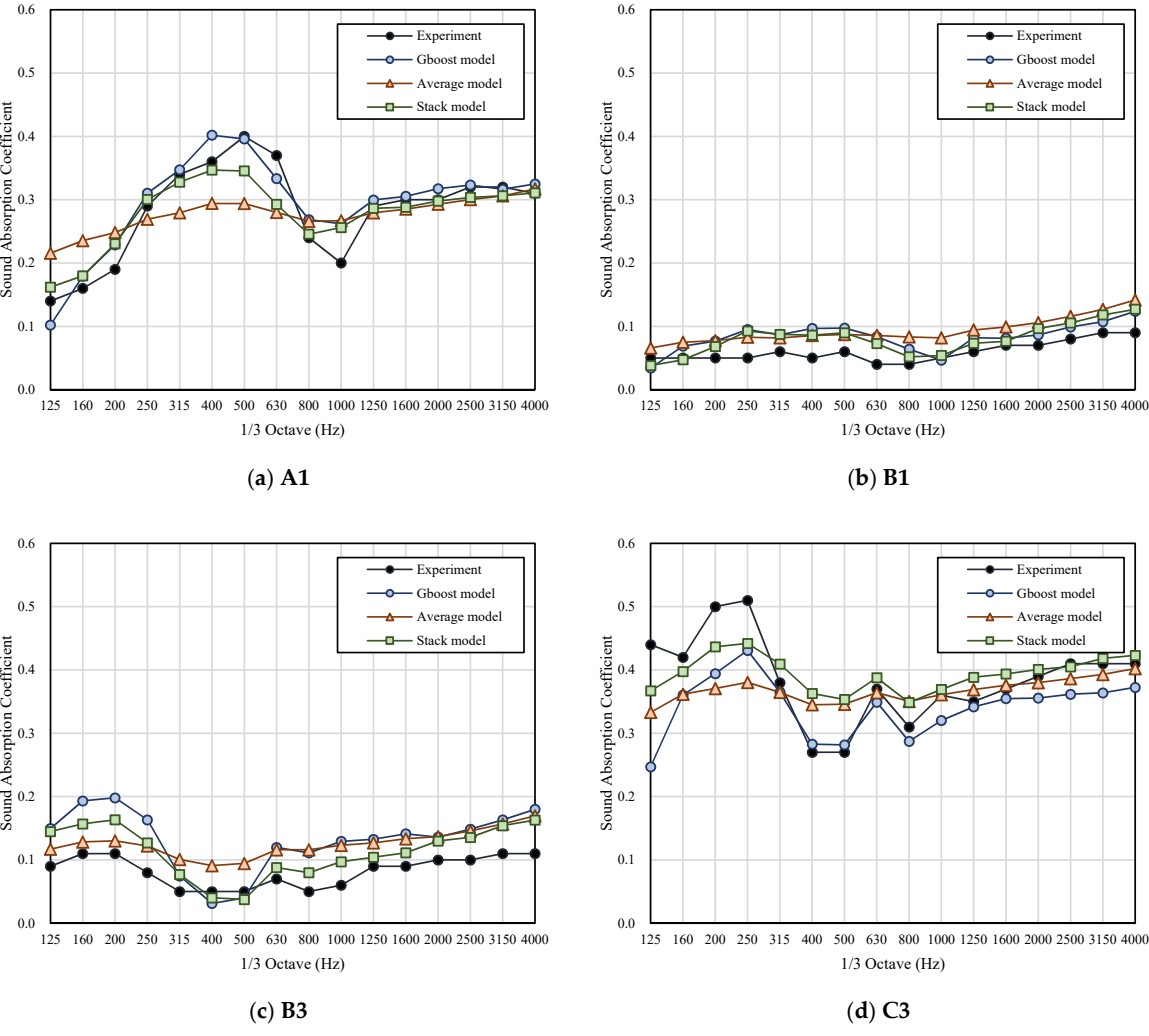

**Figure A2.** *Cont.*

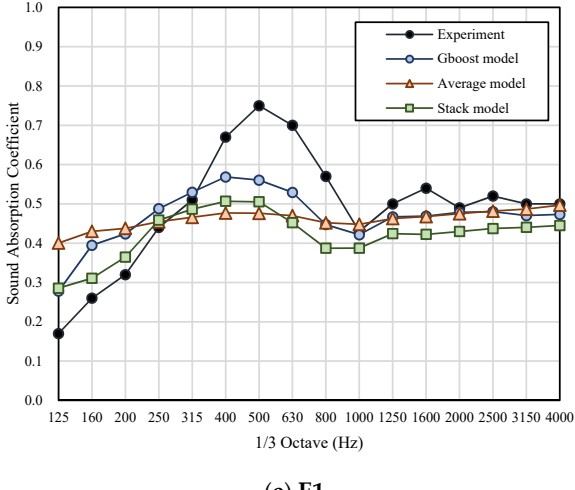

(**e**) **E1**

**Figure A2.** Prediction Results of the ML Model.

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
