# Peer review of "A Machine Learning Based Prediction Model for the Sound Absorption Coefficient of Micro-Expanded Metal Mesh (MEMM)"

_applsci, doi:10.3390/app10217612_

Round 1
Reviewer 1 Report
The topic of the paper is of undoubted interest and the results obtained are very interesting even if apply only to the examined context. Prediction models are very sought-after in acoustics especially when laboratory tests for the determination of absorption properties are expensive and require very large samples. The article is well written even if it requires a fair knowledge of Learning Machine procedures; perhaps the description part of these could be expanded.
The paper can be published after minor revision.
See detailed remarks as follows.
Detailed remarks:
Pag.2 lines 47-48: can you justify this sentence with some reference?
Pag.2 line 56: ...three machine learning models of supervised the learning method...
Pag.2 Table 1: add the meaning of the two symbols 0 and X.
Pag.4 line 79: ...specimen was 10.8 m2 (3 m × 3.6 m),...
Pag.4 line 80: (Figure 4): is there a picture of the specimen?
Pag.5 line 102: at the end of the sentence add R is the...
Pag.7 point 2.3: expand, if possible, the description of the ML method to facilitate understanding for those who have never used it.
Pag.7 line 138: ...(with k=5)
Pag.7 line 141: gradient boosting
Pag.8 line 178: also write in full root mean square error (RMSE)
Pag.10 lines 188-191: describe in more detail the results shown in the diagrams of Figure 10.
Appendix A:
add the following: R, T, D, t, γ
Author Response
Thanks for your comment, and we have revised the paper as the attached file.
- Pag.2 lines 47-48: can you justify this sentence with some reference?
Add a description to show the predictive performance of neural network models and conventional methods. (Pag. 2 line 46-49)
- Pag.2 line 56: ...three machine learning models of supervised the learning method...
Revised. (Pag. 2 line 59)
- Pag.2 Table 1: add the meaning of the two symbols 0 and X.
Add the meaning at table description. (Pag. 2 line 66)
- Pag.4 line 79: ...specimen was 10.8 m2 (3 m × 3.6 m),...
Revised. (Pag. 3 line 82)
- Pag.4 line 80: (Figure 4): is there a picture of the specimen?
Add the picture of specimen at Figure 4. (Pag. 4 line 86)
- Pag.5 line 102: at the end of the sentence add R is the...
Revised. (Pag. 5 line 110)
- Pag.7 point 2.3: expand, if possible, the description of the ML method to facilitate understanding for those who have never used it.
Add the description of the average model and the stacking model. (Pag. 8 point 2.3)
- Pag.7 line 138: ...(with k=5)
Revised. (Pag. 7 line 146)
- Pag.7 line 141: gradient boosting
Revised. (Pag. 7 line 149)
- Pag.8 line 178: also write in full root mean square error (RMSE)
Revised. (Pag. 8 line 191)
- Pag.10 lines 188-191: describe in more detail the results shown in the diagrams of Figure 10.
Add some detail about Figure 10. (Pag. 10 line 203-208)
- Appendix A: add the following: R, T, D, t, γ
Add R, T in Appendix A.
D, t, γ has been list in Appendix A(the nomenclature of the 3 conventional models have been reorganized so that the same parameter in different models has the same meaning). Also rearranged the list according to the appearance order.

Reviewer 2 Report
The paper is well written and fits the aims and scope of the journal.
The main idea of the paper is interesting and it appears that an extensive
experimental campaign was carried on. Anyway results from the measurements
can be used more efficiently in this paper.
- Figure 1 is out of focus, please provide higher resolution images if possible.
- Line 79: in "m2" the "2" must be superscript.
- The geometry of the micro expanded metal mesh is not clearly described, e.g. the perforation shape along the main axe is not clear enough: does figure 5 represent the real shape of the holes?
- 14 samples are described, but the figures only show results from 2 of them. The geometrical parameters presented in Table 2 that characterize each sample are not clear.
- The equations of the model are presented as a list of equations without a text that explain the meaning of each one and the meaning of each variable. In particular the use of Delaney-Bazley for MPP or meshes need a deeper presentation or a specific reference. The Delaney Bazley method is empirical (not "theoretical"), is superseded by Miki's one (Miki Y., Acoustical properties of porous materials - Modifications of Delany-Bazley models, J. Acoust. Soc. Jpn (E). 11(1), 1990, pp. 19-24) and is usially applied to fibrous materials such as rock wool or glass wool. For other kinds of materials, different derived formulas have been obtained in the last decades (e.g. Garai-Pompoli, Wu Qunli, Dunn-Davern, Komatsu, etc.). The use of this formulas for a metal mesh would give for sure huge errors in the prediction of the acoustical impedance, as it in fact happens in the paper. Why was this model used? Its bad performance was pretty obvious to predict.
- How were obtained the non-acoustic parameters for the material tested?
- 3 semi-theoretical models are presented in the text but only 2 of them are used for the discussion.
- The scatter plots are used to compare the accuracy of the models, it seems that in this way the machine learning models have a better performance (also if the bad performance of Delaney-Bazley was clear from the beginning), but i think that a frequency based analysis can give more precise information of the weakness of each prediction model.
- The discussion and conclusion seems more like an abstract and is lacking a more deep contrast of the results and show a possible hypothesis around why the results come in the presented way.
- In the nomenclatre, several units are missing, e.g. the propagation constant (rad/m)
Author Response
Thanks for your comment, and we have revised the paper as the attached file.

Round 2
Reviewer 2 Report
The authors improved their paper giving all the explanations needed.
Just some minor corrections:
- line 47: reference 9 is "Falcon Perez", Ricardo is the name of the author, not the surname"
- line 49: after the semicolon no uppercase letter is needed
- line 230: "large-size —reverberation rooms" sounds better than "room", because the authors used "tubes" before.